# Using high-definition transcranial direct current stimulation to investigate the role of the dorsolateral prefrontal cortex in explicit sequence learning

Hannah K. Ballard[1]*, Sydney M. Eakin[2], Ted Maldonado[2], Jessica A. Bernard[1,2]

1 Texas A&M Institute for Neuroscience, Texas A&M University, College Station, Texas, United States of America, 2 Department of Psychological and Brain Sciences, Texas A&M University, College Station, Texas, United States of America

* hannah_ballard@tamu.edu

**Data Availability Statement:** The data file used for analyses is available from the Open Science

## Abstract

Though we have a general understanding of the brain areas involved in motor sequence learning, there is more to discover about the neural mechanisms underlying skill acquisition. Skill acquisition may be subserved, in part, by interactions between the cerebellum and prefrontal cortex through a cerebello-thalamo-prefrontal network. In prior work, we investigated this network by targeting the cerebellum; here, we explored the consequence of stimulating the dorsolateral prefrontal cortex using high-definition transcranial direct current stimulation (HD-tDCS) before administering an explicit motor sequence learning paradigm. Using a mixed within- and between- subjects design, we employed anodal (n = 24) and cathodal (n = 25) HD-tDCS (relative to sham) to temporarily alter brain function and examine effects on skill acquisition. The results indicate that both anodal and cathodal prefrontal stimulation impedes motor sequence learning, relative to sham. These findings suggest an overall negative influence of active prefrontal stimulation on the acquisition of a sequential pattern of finger movements. Collectively, this provides novel insight on the role of the dorsolateral prefrontal cortex in initial skill acquisition, when cognitive processes such as working memory are used. Exploring methods that may improve motor learning is important in developing therapeutic strategies for motor-related diseases and rehabilitation.

## Introduction

Movement coordination and motor skill acquisition are important for routine functioning and day-to-day tasks (i.e., riding a bike, playing a musical instrument, and playing sports). The ability to adequately learn and perform a skill is crucial for the execution of conventional actions that impact everyday tasks. Moreover, the capacity to relearn motor skills after injury or infarct is also of interest, considering the impact of motor-related issues on daily functioning. As such, investigating the neural circuitry and underlying mechanisms of motor sequence

Framework database (DOI: 10.17605/OSF.IO/YNE5Z).

**Funding:** The author(s) received no specific funding for this work.

**Competing interests:** The authors have declared that no competing interests exist.

learning is important in advancing our current understanding of this behavior and informing protocols and therapeutic strategies for improved motor function and rehabilitation.

Skill acquisition typically develops over distinct learning phases beginning with an early, or fast, phase during the initial stages of learning and transitioning into a late, or slow, phase as automaticity develops [1]. The early learning phase is characterized by more cognitively-focused processes and requires active thinking and working memory while the skill is initially being acquired [2]. The late learning phase is more motor-focused as the skill becomes automatic through repetition and practice. Different brain areas are involved in each learning phase, assisting via their unique functions [3]. The striatum is particularly active during the late learning phase when a motor skill becomes implicit, whereas the prefrontal cortex (PFC) and cerebellum (CBLM) have been implicated in the working memory and cognitive aspects of early learning [4–8]. In fact, research suggests that the CBLM impacts cognitive processes through a closed-loop cerebello-thalamo-prefrontal circuit [9–14], which may be particularly important during initial skill acquisition. Notably, we have recently suggested that the PFC is the nexus of this circuit, governing the operations of early sequence learning, while the CBLM may serve more of a supporting role [15]. The PFC coordinates the cooperation of other brain areas involved in sequence learning [16] and is considerably engaged in the early stages of skill acquisition when cognitive and working memory resources are primarily at play [2,5,17,18].

Though we have a general understanding of the brain areas involved in sequence learning [7], there is still more to discover about the neural mechanisms underlying the acquisition of a new skill. Skill acquisition may be aided by interactions between the CBLM and PFC through the proposed cerebello-thalamo-prefrontal cognitive network [19]; however, this idea remains generally speculative. Our goal in the current study is to investigate the notion that this network, and especially the PFC, is involved in initial skill acquisition, while also shedding light on the cognitive aspects of explicit sequence learning. Thus, our current work focuses on the role of the PFC in the cognitive aspects of sequence learning, in comparison to prior work investigating the role of the CBLM [15]. Understanding the unique and relative contributions of these two areas in a key circuit will provide novel insight into the processes involved in sequence learning. In our previous research, we found polarity-specific effects of CBLM stimulation on the acquisition of a new motor skill. Anodal tDCS negatively impacted sequence learning and cathodal tDCS had little impact on performance [15]. Here, we wanted to replicate the same experimental design with the PFC. Ultimately, we aim to better understand the neural underpinnings of sequence learning, particularly with respect to the involvement of cognitive systems and their associated networks.

To this end, we can investigate the role of certain brain areas and probe the underlying circuitry of sequence learning with transcranial direct current stimulation (tDCS). tDCS involves applying small amounts of electrical current to the scalp with an array of electrodes that can target a specific brain area and temporarily alter its function [20]. tDCS impacts the firing rates of neurons and can either increase or decrease neuronal activity in the targeted area, depending on the type of stimulation administered (cathodal or anodal) [21,22]. This technique of non-invasive brain stimulation has been successful in temporarily modulating cognition and motor learning in multiple experimental settings [23–25]. Research using stimulation of the dorsolateral PFC (DLPFC) has demonstrated an influence of tDCS on the cognitive aspects of motor learning [26–31], likely due to the impact on working memory processes that provide essential contributions to skill acquisition. Other work has shown no impact of DLPFC stimulation on sequence learning or consolidation [32–35]; however, these studies specifically evaluated implicit learning. On the contrary, work by Greeley and Seidler [36] presented evidence for improved implicit sequence learning after anodal stimulation to the left DLPFC, while right DLPFC stimulation impeded learning. Additionally, Talimhkani et al [37]

found improvements in retention with anodal tDCS to the left DLPFC, but no differences in skill acquisition were observed. Generally speaking, however, to this point results of these studies using tDCS to investigate the DLPFC during learning have been mixed. Notably, the majority of these investigations use the traditional 2-pad stimulation technique [32–34,36,37], which involves applying greater amounts of electrical current to larger cortical areas, thereby delivering diffuse stimulation to the brain, and in turn, what is likely a more distributed impact on brain activity.

In an effort to improve targeting with tDCS, we implemented high-definition tDCS (HD-tDCS) in the current study. This advanced method achieves increased focality compared to the traditional 2-pad stimulation technique by distributing current over multiple electrodes. Though some recent studies have taken advantage of HD-tDCS to evaluate cognitive performance [38–40], more work is needed to fully comprehend the capacity with which HD-tDCS impacts activity in the PFC with particular respect to explicit motor sequence learning. Indeed, the improved targeting that is possible with this approach stands to provide further insight into the role of the DLPFC in the cognitive components of sequence learning. Here, we targeted the left DLPFC to get at the working memory processes and cognitive aspects of explicit motor sequence learning, using both anodal and cathodal HD-tDCS. We chose this particular location for its role in the control of cognition and working memory, specifically in the context of sequence learning, as several investigations using functional magnetic resonance imaging (fMRI) and positron emission tomography (PET) have indicated activation in the DLPFC during the early stages of skill acquisition [18,41–43]. In line with results from our previously collected data on the CBLM, we predicted differing impacts of stimulation type (cathodal vs. anodal), as well as stimulation condition (active vs. sham), on sequence learning performance.

## Materials and methods

### Participants

This study was carried out on two distinct groups of individuals, resulting in two completely independent samples. Each individual was screened for exclusion criteria associated with tDCS [20] in order to ensure the safety of those that participated. Both groups underwent the same experimental procedures but differed in the type of stimulation received (cathodal vs. anodal). The first group consisted of twenty-five individuals that underwent cathodal and sham stimulation during two separate HD-tDCS sessions, counterbalanced in order. Though data was collected for thirty-five subjects, one individual was discontinued due to discomfort from stimulation and four individuals did not return for the second session of the study. Additionally, we applied an *a priori* exclusion criterion based on behavioral performance to avoid the influence of outliers, such that individuals with performance 3 standard deviations below the group mean were completely excluded from all analyses. In the cathodal sample, this resulted in the exclusion of five subjects that had only 34% accuracy, or less, across trials. Such low accuracy levels indicate that these participants were not adequately performing the task with undivided attention, and may have lacked motivation, thereby resulting in the exclusion of their full data from analyses. Thus, the final sample for the first group consisted of twenty-five healthy young adults (15 females, mean age 19.2 years ± 0.9 S.D., range 18–21 years).

The second group consisted of twenty-four individuals that also underwent two HD-tDCS sessions; however, in this group, anodal and sham stimulation were administered in a counterbalanced order. Forty-three subjects were initially enrolled in the study. Five individuals were excluded due to our exclusionary criteria to ensure participant safety, and two individuals discontinued stimulation due to discomfort. Further, one individual was excluded in light of technical difficulties with software updates to the testing computer and five did not return for the

second session. After applying the same *a priori* cutoff to exclude behavioral outliers, six additional individuals were completely excluded from analyses due to accuracy scores of 33% or less, which indicates that these subjects may not have been paying attention. These exclusions left twenty-four healthy young adults in the final sample for the second group (11 females, mean age 18.7 years ± 0.6 S.D., range 18–20 years).

None of the subjects included in the final sample for either group had any history of neurological disease (e.g., epilepsy or stroke) nor a formal diagnosis of psychiatric illness (e.g., depression or anxiety), and, relatedly, none were taking medications that could potentially interfere with central nervous system functioning (e.g., neuroleptics, narcotics, anxiolytics, analgesics, stimulants, or antidepressants). All subjects were right-handed in both the cathodal group (mean score 92.5 ± 2.2 S.D.) and anodal group (mean score 81.2 ± 26.6 S.D.), as assessed by the Edinburgh Handedness Inventory [44]. Subjects received partial course credit for their participation and were recruited through the Psychology Subject Pool at Texas A&M University. Participation in the current study was limited to individuals that had not previously participated in a study using tDCS and were, thereby, tDCS naïve. All subjects provided written informed consent before study procedures began, and our protocol was approved by the Institutional Review Board at Texas A&M University.

## Procedure

**HD-tDCS.** In each group, participants underwent two HD-tDCS sessions, separated by exactly one week, and received both active (cathodal for the first group and anodal for the second) and sham stimulation in a mixed within- and between- subjects design. In order to keep all subjects blind to study procedures, participants were not informed of which stimulation condition was administered during each session until the debriefing period at the end of the second session. The order of stimulation sessions (active or sham) was counterbalanced across all subjects in both groups. Regardless of stimulation group, tDCS was administered with a Soterix MxN HD-tDCS system (Soterix Inc., New York, NY) and an 8-electrode montage. The electrode array and corresponding current intensities (Table 1), as well as the modeled current flow (Fig 1) for DLPFC stimulation, were obtained through HD-Targets software. This software allows for stimulation of a targeted region of interest with increased focality [45,46], as compared to the traditional 2-electrode tDCS system. In order to effectively deliver 2.0 mA of net current with the acquired montage, different intensities were implemented for each

**Table 1. Stimulation parameters.**

|   | Location | Current |
|---|----------|---------|
| 1 | FP1 | 1.0062 |
| 2 | F3 | -1.1757 |
| 3 | F4 | 0.2221 |
| 4 | P3 | 0.0761 |
| 5 | C2 | 0.0183 |
| 6 | CP2 | 0.1049 |
| 7 | F5 | -0.8243 |
| 8 | TP7 | 0.0699 |
| C | F9 | 0.5024 |

Electrode array and current intensities for prefrontal stimulation. *Location*: particular brain locations for electrode placement, determined by 10–20 system; *Current*: specific intensity of electrical current (mA) applied at a given location.

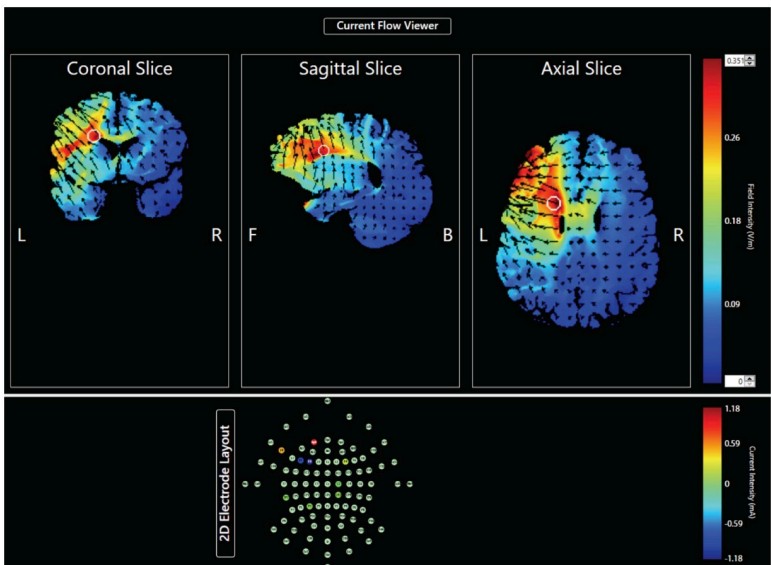

**Fig 1. Modeled current flow for prefrontal stimulation (BA 46) using HD-Targets software.** Color bar indicates field intensity (V/m) on a scale of 0–0.35 (bottom to top). *L*: Left; *R*: Right; *F*: Front; *B*: Back.

individual electrode location. Table 1 and Fig 1 depict this information for cathodal DLPFC stimulation specifically; however, the same magnitude of intensities and modeled current flow are used for the anodal montage, with the sign of each current intensity flipped and the opposite direction of current flow. For the cathodal group, we used a "spiral out" approach, whereas a "spiral in" approach was used for the anodal group, which simply refers to the direction of current flow for a specific type of stimulation when using a montage that contains multiple stimulation electrodes.

During each HD-tDCS session, and for both experimental groups, 0.1 mA was initially delivered for one minute to establish a sufficient connection, which is key for effective stimulation. The connection was considered adequate once a resistance level of 50 kilo-ohms or lower was achieved at each individual electrode. Following this pre-stimulation period, the full stimulation session began with a gradual increase in current that occurred for roughly 30 seconds until the intended intensity of 2.0 mA was reached. At this point, current remained steady for 20 minutes if the subject was assigned to the active stimulation condition (cathodal or anodal). However, for the sham session, current gradually decreased and returned to zero for the remainder of the stimulation period before ramping back up and then down again during the last 30 seconds. Regardless of the assigned condition, subjects experienced the same standard setup for HD-tDCS during both sessions and were blinded to the condition being administered for each session. We specifically delivered offline stimulation in this investigation, as stimulation approach (offline vs. online) has been shown to impact the efficacy of tDCS [47]. Importantly, in the context of our work here, a meta-analysis by Hill, Fitzgerald, and Hoy [48] noted working memory improvements for offline tasks, but not online, after anodal tDCS. As such, we chose to use offline stimulation in consideration of these differences, and in order to be consistent with our prior work [15].

**Sequence learning.** We used an explicit motor sequence learning paradigm, modeled directly after Kwak et al [49] and used in our past work [15], in order to assess the impact of DLPFC stimulation on the ability to acquire a motor skill and learn a new pattern of events. As research using fMRI has indicated more activation in brain areas associated with cognitive

processing during explicit learning, compared to implicit [18,42,50,51], we chose to investigate the influence of stimulation with an explicit sequence learning paradigm to explore its effects on the cognitive, rather than solely motor, aspects of skill acquisition.

This behavioral task was administered during both experimental sessions for each group following the 20-minute tDCS period and took roughly 12 minutes to complete. The task was displayed on a computer screen using Presentation Software (2019 Neurobehavioral Systems, Inc.) and participants were instructed to place their left middle, left index, right index, and right middle fingers on the numbers 1, 2, 3, and 4 of the computer keyboard, respectively. The task was designed to be bimanual, as is common in sequence learning research [17,49,52–55], in order to allow for potential translation to an MRI environment for future work. During the task, four white rectangles outlined in black were presented on the screen, and these rectangle locations corresponded to the instructed finger placements from left to right. A single rectangle was shaded to black during each individual trial, and participants were instructed to press the appropriate button matching the location of that rectangle as quickly and accurately as possible. Each stimulus was displayed for 200 milliseconds and participants were given 800 milliseconds to make a response before the next stimulus appeared. For each trial, a different rectangle location was shaded to black. The task consisted of 15 total blocks, with some being random and some containing a repetitive sequence of events. Random blocks were dispersed throughout the sequence blocks in order to account for baseline motor function, and, as the task was intended to be clearly explicit, each block was preceded by either an "R" or an "S", depending on the nature of the upcoming trials.

Relatedly, our sequence learning paradigm did not include extensive practice or follow-up retention tests and was relatively short in length to further maintain its explicit character. As such, each participant completed the following block design during both sessions: R-S-S-S-R-R-S-S-S-R-R-S-S-S-R, amounting to 6 random and 9 sequence blocks overall [49]. Each random block contained 18 trials while each sequence block contained 36 trials, totaling to 432 trials [49]. During the sequence blocks, a 6-element pattern was repeated throughout the task, and this pattern was distinct for each experimental session in order to avoid any interference from practice effects. The same pattern was presented for each participant during the first session (2-4-1-3-2-4), throughout the entire task, and a different pattern was used for the second session (3-2-4-1-2-4). Thus, the same sequence was repeated across sequence blocks for each session, but sequences differed between the two experimental sessions. These sequences did not differ in complexity according to the Kolmogorov-Smirnov test ($D = 0.184$, $p = 0.346$), and neither contained any sequential numbers or trills (e.g., 1-2-1-2).

**Debriefing.** After subjects completed both sessions of the experiment, they were asked to answer a series of questions regarding sensations experienced during the stimulation sessions, such as itching, pain, or burning. Responses from this questionnaire were used to determine whether the sensations experienced from stimulation may have interfered with subjects' ability to subsequently perform the task. Results indicate that stimulation sensations were not considerably strong, or thereby distracting, across groups and sessions (Table 2). After the second session, subjects were also asked to indicate which stimulation condition (active or sham) they received during each session in order to gauge the effectiveness of our sham technique. These indications were made based off of the subjective experience during the two sessions, and subjects were given the options of "real", "placebo", or "I don't know" for each session. Responses of "I don't know" were coded as incorrect. When reporting on the stimulation condition received, 50% of subjects that received active stimulation during the first session guessed correctly, while only 21.74% of those that received sham stimulation correctly identified their stimulation condition. When asked to report what was received for the second session, of those that received active stimulation in the second session, 60.87% were correct in identifying

**Table 2. Stimulation sensations questionnaire.**

| Sensation | Mean | Standard Deviation |
|---|---|---|
| Itching | 1.73 | 1.11 |
| Pain | 0.67 | 0.85 |
| Burning | 0.88 | 0.86 |
| Warmth/Heat | 0.51 | 0.71 |
| Pinching | 0.47 | 0.65 |
| Metallic/Iron Taste | 0 | 0 |
| Fatigue | 0.69 | 1.14 |

Means and standard deviations for each sensation experienced during stimulation, rated on a scale of 1 to 5, across groups and sessions.

the stimulation condition, while those that received sham stimulation were correct only 38.46% of the time. In general, individuals were better at identifying active stimulation, but at best, they were only marginally better than chance, suggesting that all participants were effectively blinded to the stimulation conditions. Lastly, subjects were debriefed on the study purpose, and the order of stimulation conditions across sessions was disclosed. A broad overview of our full experimental procedure is depicted in Fig 2.

## Data analysis

**Accuracy.** Behavioral performance was primarily assessed using average total accuracy (ACC) per block on the sequence trials, specifically. ACC scores were calculated as the percent of correct button presses out of the total number of trials for each sequence block. Correct, incorrect (e.g., commission errors), and missed (e.g., omission errors) responses were included in the final number for total trials. Each triplet of sequence blocks was concatenated into one respective learning phase, creating three subsequent learning phases denoted by early, middle, and late. This *a priori* step was done in order to replicate the primary behavioral measure used in Kwak et al [49] and in the interest of considering timing-specific effects of stimulation on motor sequence learning. This approach allows us to investigate the broader changes in skill acquisition over the course of the learning task as our paradigm was relatively brief with only 9 sequence blocks total. Rather than modeling ACC over each individual sequence block, we chose to simplify and streamline our analyses in the effort of avoiding redundant comparisons that could potentially obscure the interpretation of results. This particular approach, in the context of ACC during a sequence learning paradigm, has been similarly adopted in prior studies of the same scope [17,49,54], including our own recent work [15]. Nonetheless, our experimental design for behavioral performance reflects an overall early learning paradigm as the task is fairly short in length, lasting only 12 minutes on average. The task was deliberately devised to evaluate initial skill acquisition as we were primarily interested in the cognitive

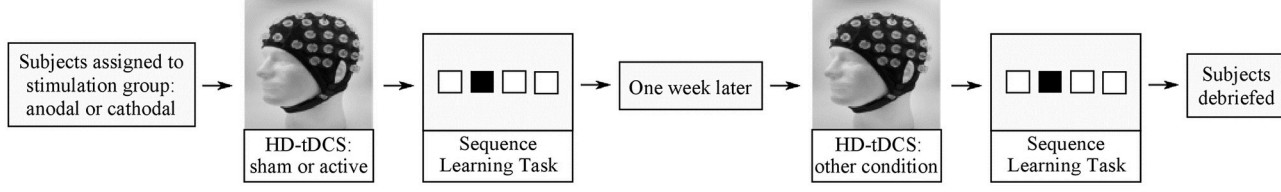

**Fig 2. Overview of experimental procedure.** HD-tDCS image source: Soterix Medical.

components of sequence learning for the current study. Further, ACC is expected to change over the course of the task for sequence blocks as participants become more familiar with the repeated pattern, while ACC during random blocks generally remains relatively consistent over time. As such, we chose to focus only on sequence blocks for this particular performance measure.

**Reaction time.** We used mean reaction time (RT) for correct responses as an additional measure of behavioral performance, considering both random and sequence trials for this variable. Comparing RT between block types allows us to confirm that we have successfully implemented an explicit learning paradigm by evaluating RT differences in the context of the broader sequence learning literature. Trials in which the participant did not respond in time, or missed, were excluded. Only those trials in which a correct response was made during a random or sequence block were used to calculate RT averages.

**Statistical models.** In order to quantify any potential differences in performance between stimulation groups (cathodal vs. anodal) and stimulation sessions (active vs. sham), mixed model ANOVAs were performed for both ACC and RT. Stimulation group was treated as a between-subjects variable, and stimulation session as a within-subjects variable. All analyses were *a priori* and performed using the ezANOVA and t.test packages in R Programming Software, with a threshold for significance of 0.05. Multiple comparisons corrections were conducted with the Bonferroni method for all follow-up analyses, using the p.adjust package in R Programming Software.

## Results

### Accuracy

A 2 x 2 x 3 mixed model ANOVA (stimulation group by stimulation session by learning phase) revealed a significant main effect of learning phase on ACC for sequence blocks, $F(2, 94) = 15.67$, $p < .001$, $\eta^2 = 0.049$, but did not reveal a significant effect of stimulation group (cathodal vs. anodal), $F(1, 47) = 0.21$, $p = 0.652$, $\eta^2 = 0.003$, or stimulation session (active vs. sham), $F(1, 47) = 1.47$, $p = 0.232$, $\eta^2 = 0.003$. This model also revealed a significant interaction between learning phase and stimulation group, $F(2, 94) = 5.07$, $p = 0.008$, $\eta^2 = 0.016$, as well as an interaction between learning phase and stimulation session, $F(2, 94) = 3.87$, $p = 0.024$, $\eta^2 = 0.009$. All remaining interactions were non-significant (all $ps > .05$). Across groups and sessions, performance changes over the course of the task; however, the nature of this change may depend, in part, on the stimulation group (cathodal vs. anodal) or the stimulation session (active vs. sham) (Fig 3A).

Though the three-way interaction was not significant, we then conducted follow-up comparisons in order to better understand these other two interactions (learning phase and both stimulation session and group), using 2 x 2 mixed model ANOVAs (stimulation group by stimulation session) on each individual learning phase. While all effects and interactions were non-significant for the early and middle learning phases (all $ps > 0.105$), the effect of stimulation session (active vs. sham) on ACC in the late learning phase was trending towards significance, $F(1, 47) = 5.86$, $p_{adj} = 0.058$, $\eta^2 = 0.023$, after applying a Bonferroni multiple comparisons correction. Further, there was no significant effect of stimulation group (cathodal vs. anodal) on ACC in the late phase, $F(1, 47) = 2.16$, $p_{adj} = 0.446$, $\eta^2 = 0.036$, nor an interaction between stimulation group and stimulation session, $F(1, 47) = 2.43$, $p_{adj} = 0.377$, $\eta^2 = 0.010$, both after correcting for multiple comparisons. These results reveal a potential trend related to stimulation session (active vs. sham) on sequence learning performance in the later stages of the task, though it does not reach the standard threshold for significance when using a conservative multiple comparisons correction approach. As depicted in Fig 3B, it seems that, on

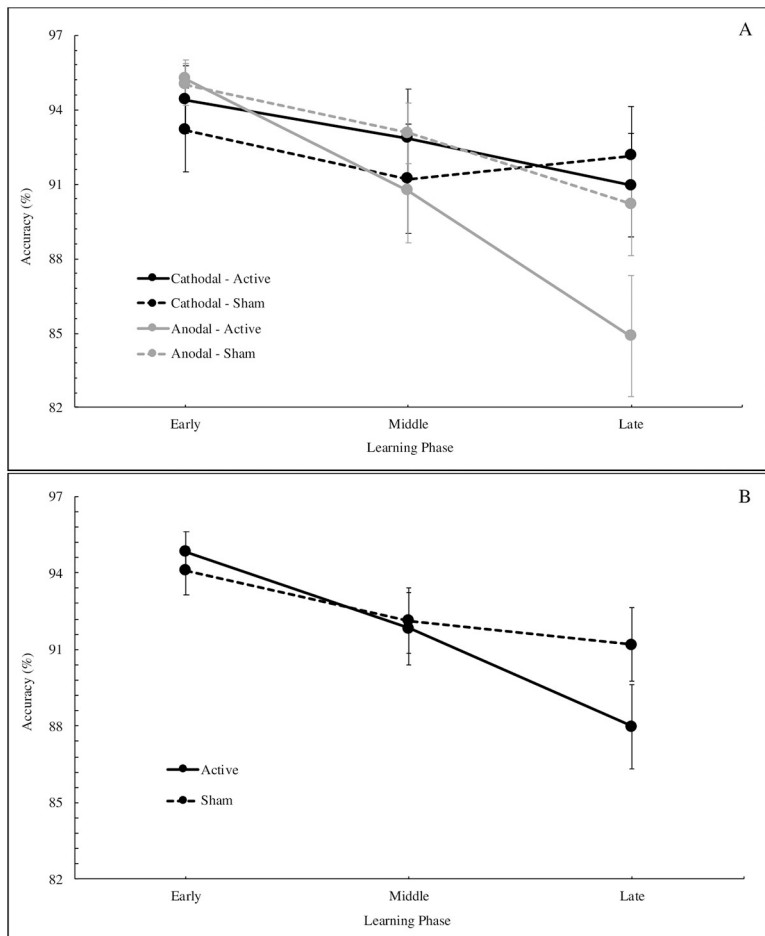

**Fig 3.** A: Accuracy across sequence learning phases by prefrontal stimulation group (Cathodal vs. Anodal) and session (Active vs. Sham). B: Accuracy by stimulation session, groups combined. Error bars indicate standard error (SE).

average, active PFC stimulation results in lower ACC scores on late sequence learning trials, compared to sham, and this negative impact on performance is revealed across both anodal and cathodal groups.

## Reaction time

We then looked at stimulation effects on RT across both random and sequence blocks by conducting a 2 x 2 x 2 mixed model ANOVA (stimulation session by stimulation group by block type). This analysis revealed a significant effect of block type (random vs. sequence), $F (1, 47) = 201.62$, $p < .001$, $\eta^2 = 0.364$, and a significant effect of stimulation session (active vs. sham), $F (1, 47) = 4.62$, $p = 0.037$, $\eta^2 = 0.007$, on RT. In addition, there was a significant interaction between block type, stimulation session, and stimulation group (cathodal vs. anodal), $F (1, 47) = 7.61$, $p = 0.008$, $\eta^2 = 0.007$; however, all remaining effects and interactions were non-significant (all $ps > 0.115$). These results indicate that RT differs between block types, demonstrating that subjects performed the task more quickly, as indicated by lower RTs, on sequence blocks compared to random (Fig 4A). Further, RT is significantly impacted by stimulation session (active vs. sham) across both block types, and the three-way

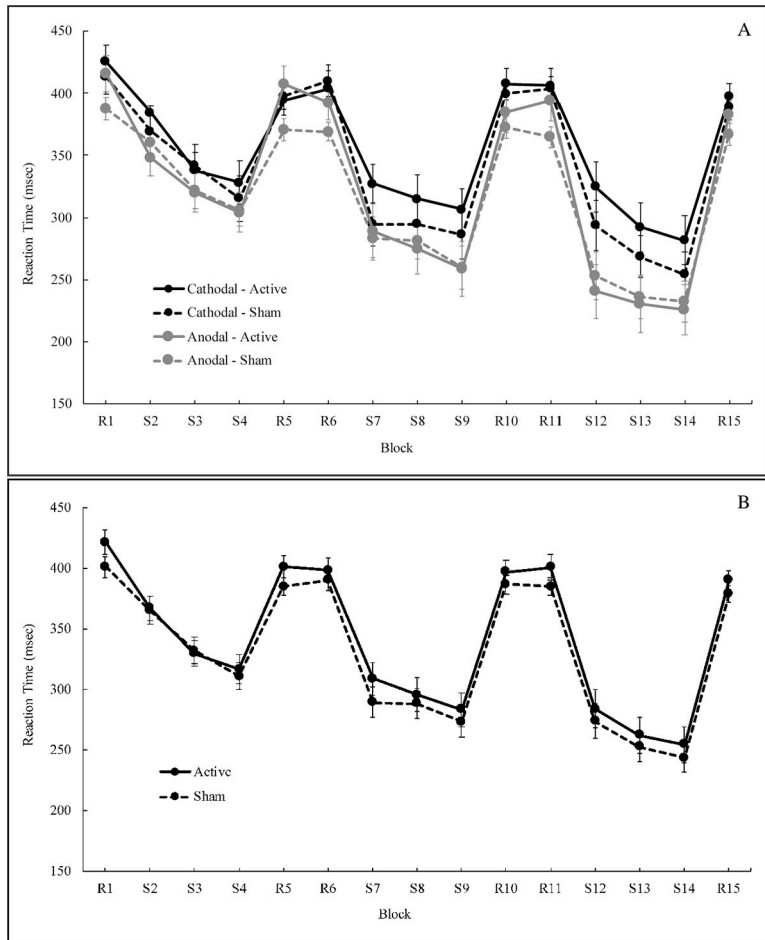

**Fig 4.** A: Reaction time across all blocks by prefrontal stimulation group (Cathodal vs. Anodal) and session (Active vs. Sham). B: Reaction time by stimulation session, groups combined. Error bars indicate SE.

interaction suggests that stimulation group (cathodal vs. anodal) may also further influence reaction time.

To investigate these results further, we then conducted 2 x 2 mixed model ANOVAs (stimulation group by stimulation session) on random and sequence blocks separately. Though no significant effects emerged when considering RT during sequence blocks (all $ps > 0.114$), a significant effect of stimulation session (active vs. sham) was present for the model conducted on random blocks, $F(1, 47) = 7.42$, $p_{adj} = 0.018$, $\eta^2 = 0.018$, after applying a Bonferroni correction. However, there was not a significant main effect of stimulation group (cathodal vs. anodal) on RT for random blocks, $F(1, 47) = 2.03$, $p_{adj} = 0.322$, $\eta^2 = 0.037$, after multiple comparisons correction. Notably, the interaction between stimulation session and stimulation group was not significant for this model, $F(1, 47) = 4.15$, $p_{adj} = 0.095$, $\eta^2 = 0.010$. This suggests that during random blocks, in particular, RT is significantly impacted by stimulation session (active vs. sham), seemingly irrespective of stimulation group (cathodal vs. anodal). Specifically, we saw slower RTs on random blocks after active stimulation (regardless of stimulation group), compared to sham (Fig 4B), which suggests that PFC stimulation has an overall negative influence on RT, and this may extend to performance more generally.

## Discussion

### Summary

Applying HD-tDCS to the left DLPFC significantly influenced performance on an explicit motor sequence learning paradigm. Stimulation impacted sequence learning over the course of the task as performance differences emerged in the middle to late learning phases. Additionally, the nature of the observed effects was dependent upon the stimulation session; active PFC stimulation had an overall negative impact on task performance, regardless of polarity, compared to sham. After active stimulation to the PFC, we observed decreases in ACC during sequence blocks (Fig 3B), though this effect did not remain significant after correcting for multiple comparisons. In addition, we observed higher RT during random blocks (Fig 4B) after active stimulation, compared to sham, and this effect remained significant after corrections. Overall, these observations are indicative of performance declines after both anodal and cathodal DLPFC stimulation with HD-tDCS. Research on cathodal tDCS is generally inconclusive and largely suggests that this type of stimulation does not uniformly influence cognitive performance [56–60]. Our results from the current study broadly support this notion with respect to the PFC. Conversely, anodal tDCS is more consistently thought to benefit behavioral performance as a result of increasing firing rates in the targeted brain area [21,22,25,27]; however, here, we observed the opposite effect on behavior.

### Prefrontal stimulation and explicit motor sequence learning

When comparing across PFC stimulation groups (cathodal vs. anodal) and stimulation sessions (active vs. sham), we found that sequence learning ACC significantly changed over the course of the task. In addition, stimulation differences emerged towards the final stages of the task. More specifically, we found that active PFC stimulation negatively impacts performance in the late learning phase, compared to sham, though this effect was not significant after correcting for multiple comparisons. ACC decreases between the early and late learning phases by about seven percent after active stimulation, whereas sham stimulation yields relatively small changes in performance with only a three percent decrease between early and late phases. Importantly, these effects are observed across both anodal and cathodal groups, indicating that stimulation polarity does not significantly influence explicit motor sequence learning when applying tDCS to the DLPFC. Any HD-tDCS stimulation over the DLPFC in this investigation had a negative impact on performance.

With respect to RT, we found that subjects performed the task more quickly on sequence blocks compared to random, regardless of the stimulation parameters. Notably, these results are consistent with our prior work on the CBLM where faster RTs were observed on sequence blocks, compared to random, after both stimulation types [15], and with other past studies of sequence learning [52,54,61,62]. Unique to the current study, we found that individuals were slower after active PFC stimulation, compared to sham, on the random blocks. We speculate that this negative effect of active stimulation to the DLPFC on RT during random blocks could be, in part, the result of an impact on motor-related processes or stimulus-response preparation functions, as the DLPFC maintains connections with the primary motor cortex (M1) [63]. Importantly, this finding further supports the notion that PFC stimulation produces an overall worsening of performance as baseline motor function decreases, in conjunction with declines in sequence ACC, as previously noted.

Notably, our ACC and RT results differ in the type of block (random vs. sequence) impacted by stimulation. That is, stimulation influences ACC during sequence blocks, and RT during random blocks. This distinction is important to note as it suggests that the impact of

HD-tDCS to the DLPFC may manifest differently for specific performance measures, potentially contributing to the observed changes in behavior after stimulation. In fact, though DLPFC stimulation typically improves working memory performance [25,27,64], results are often specific to a distinct performance measure and may be more limited than presumed. Several studies have demonstrated that stimulation to the DLPFC primarily impacts performance speed, but not necessarily accuracy [28,30], even when using the HD-tDCS approach [65], which is generally consistent with our results. In some of this research, working memory improvements are selectively observed when cognitive demand is particularly high [29,30]. However, additional work reports null effects of HD-tDCS to the DLPFC on working memory performance [40,66], as well as implicit learning and consolidation [32–35]. In sum, inconsistent results have been observed with DLPFC stimulation and, though the effects observed in the current study are somewhat unexpected, our findings do not completely conflict with the literature more broadly and add to the ongoing conversations regarding the effects of prefrontal tDCS on skill acquisition.

Though unexpected in the greater context of tDCS research [21,22,25,27], our unique results could also be a product of the HD-tDCS approach employed here, as the majority of motor skill acquisition work to date relies upon the more traditional 2-pad stimulation technique [23]. While both HD-tDCS and the 2-pad technique serve to increase or decrease neural activity in the targeted brain area, purportedly due to similar molecular mechanisms, HD-tDCS is suggested to be a more effective approach as it allows for greater precision in targeting [67]. The traditional 2-pad technique, however, is limited to larger, more diffuse cortical areas. Further, these differences between stimulation approaches (HD-tDCS vs. 2-pad) manifest behaviorally as well. Work by Cole et al [68] demonstrates motor learning deficits after both active and sham HD-tDCS to right M1, but not after 2-pad M1 stimulation. Though their stimulation target was different than our own, and deficits were specific to consolidation of learned motor skills, this work reveals negative impacts of HD-tDCS on motor learning after active stimulation. While our predictions were guided by traditional tDCS research, this ultimately suggests that the results observed in the current study may not be entirely inconsistent with the limited work on the behavioral effects of HD-tDCS. Exploring this cutting-edge technique and challenging the theories constructed from work using the traditional 2-pad method is crucial to advancing the field of neuromodulation.

In general, the unexpected nature of our findings may partially result from other methodological differences, as stimulation parameters can impact the effects of tDCS on behavior [56]. For instance, whether stimulation is applied during task performance or entirely before is an important consideration as some of the excitatory effects of anodal tDCS and null effects of cathodal tDCS have been observed with online protocols [27]. Further, the body of work on the effects of DLPFC stimulation on implicit sequence learning and consolidation is also chiefly reliant on studies that used an online tDCS approach [34–37]. In the current study, we employed offline stimulation. And again, we used a more novel tDCS approach, which may have impacted brain activity and behavioral performance differently in comparison to the traditional 2-pad technique. Importantly, individual differences in responsiveness to stimulation have also been implicated in the effect on behavioral performance [60], and this inter-individual variability with tDCS is not yet fully understood [23]. Finally, a meta-analytical review by Jacobson, Koslowsky, and Lavidor [57] suggests that, though excitatory effects with anodal stimulation and inhibitory effects with cathodal are consistently observed in research examining motor function, the dual-polarity theory of anodal-excitation and cathodal-inhibition is less consistently upheld in work concerning cognitive performance. Given our interest in cognitive processes during learning, and the target of stimulation, this may be especially pertinent here.

The current work was conducted to further elucidate the role of the dorsolateral prefrontal cortex in the cognitive stages of initial skill acquisition. However, we cannot discount the input of other brain areas and circuits associated with motor sequence learning. The basal ganglia have also been implicated in sequence learning [69–72], and may contribute to the behavioral effects we observed in the current study. The cortico-striatal circuit is involved in early learning and remains particularly important after automaticity develops [69,70], which could partially come into play with our explicit learning paradigm. Further, premotor areas drive the consolidation process that occurs between prefrontal and motor cortices as a sequence transitions from explicit to automatic nature [43,69,73]. Thus, stimulating the DLPFC could facilitate increased engagement of this circuit as well. In addition, it is possible that stimulation may simply introduce random noise in the targeted brain area, or its affiliated circuits, causing an overall disruptive effect on behavioral performance. This hypothesis could help explain the lack of polarity-specific effects observed in the current study, as we found that both anodal and cathodal DLPFC stimulation disrupts performance.

## Limitations

While our findings offer new insights into the neural mechanisms of explicit sequence learning, a few limitations are worth noting. First, though HD-tDCS allows for improved targeting and increased focality, it is possible that stimulation reached areas other than the intended location (Fig 1). We suggest that our stimulation montage primarily impacted the DLPFC, however current may have spread to other regions of the PFC, including premotor or even motor cortical regions. Relatedly, we did not include a negative stimulation site control using a brain area outside of our network of interest. Though the current study was aimed at investigating a component of the cerebello-thalamo-prefrontal network and its role in motor sequence learning, future work may benefit from the inclusion of a negative stimulation site to better understand the spatial specificity of these results. In addition, our results are specific to immediate learning effects as we used an explicit paradigm, and we did not include retention tests to evaluate consolidation over time. As the long-term effects of tDCS are not clear, future work extending past our brief learning paradigm to include assessments of retention over time would be beneficial in defining the lasting impact of stimulation targeting the DLPFC on motor sequence learning. Finally, as we used an undergraduate pool to recruit subjects, motivation levels likely varied between participants in our sample as they were only required to show up in order to receive course credit. This resulted in behavioral exclusions for both experimental groups. As such, our substantial loss of subjects through behavioral exclusions, as well as attrition, may have potentially impacted results.

## Conclusions

As the current body of literature concerning the effects of PFC stimulation on motor sequence learning heavily relies on work investigating implicit aspects of skill acquisition using the traditional 2-pad technique in an online context, our work provides essential insight on the impact of offline HD-tDCS when applied to the left DLPFC. Together, our findings offer a novel perspective on the neural mechanisms underlying the cognitive aspects of explicit motor sequence learning, an area that remains relatively unexplored. Results suggest that active stimulation to the DLPFC, regardless of polarity, negatively impacts initial skill acquisition, compared to sham. The conclusions drawn here contribute to the growing body of literature that disputes the dual-polarity theory of anodal-excitation and cathodal-inhibition. Given the complicated nature of this methodology, the additional evidence provided from our work using HD-tDCS to the DLPFC is necessary to form solid conclusions. Further, effects of applying HD-tDCS to

this area emerge over the course of a brief sequence learning paradigm, potentially implicating the cerebello-thalamo-prefrontal network in the early stages of skill acquisition. Exploring methods that may improve the acquisition of essential motor skills is crucial in developing therapeutic strategies for motor-related diseases and recovery from injury, as well as reducing normative declines in motor functioning with advanced age.

## Author Contributions

**Conceptualization:** Jessica A. Bernard.

**Data curation:** Hannah K. Ballard, Sydney M. Eakin, Ted Maldonado.

**Formal analysis:** Hannah K. Ballard.

**Resources:** Jessica A. Bernard.

**Supervision:** Sydney M. Eakin, Ted Maldonado, Jessica A. Bernard.

**Visualization:** Hannah K. Ballard.

**Writing – original draft:** Hannah K. Ballard.

**Writing – review & editing:** Hannah K. Ballard, Jessica A. Bernard.

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
