## [Decision Letter · Decision Letter 0]

16 Oct 2020

PONE-D-20-29379

Using high-definition transcranial direct current stimulation to investigate the role of the dorsolateral prefrontal cortex in explicit sequence learning

PLOS ONE

Dear Dr. Ballard,

Thank you for submitting your manuscript to PLOS ONE. After careful consideration, we feel that it has merit but does not fully meet PLOS ONE’s publication criteria as it currently stands. Therefore, we invite you to submit a revised version of the manuscript that addresses the points raised during the review process.

As you will see below, the experts who have reviewed your paper have expressed significant concerns regarding the methodology and the research design. In particular, both Reviewers were concerned about the lack of specificity between anodal and cathodal stimulation. The issue of spatial selectivity is also a significant issue. I share their concerns and ask you to consider adding an experiment to address the spatial selectivity issue. There are also other considerations raised by Reviewer 2 regarding the statistical analysis that will require your attention.

We look forward to receiving your revised manuscript.

Kind regards,

François Tremblay, PhD

Academic Editor

PLOS ONE

Journal Requirements:

2. We note that Figure 2 includes an image of a participant in the study. 

Reviewers' comments:

Reviewer's Responses to Questions

**Comments to the Author**

1. Is the manuscript technically sound, and do the data support the conclusions?

Reviewer #1: Partly

Reviewer #2: Partly

2. Has the statistical analysis been performed appropriately and rigorously? 

Reviewer #1: Yes

Reviewer #2: No

3. Have the authors made all data underlying the findings in their manuscript fully available?

Reviewer #1: Yes

Reviewer #2: Yes

4. Is the manuscript presented in an intelligible fashion and written in standard English?

Reviewer #1: Yes

Reviewer #2: Yes

5. Review Comments to the Author

Reviewer #1: The authors stimulated the dorsolateral prefrontal cortex by high-definition transcranial direct current stimulation to study its role in explicit motor sequence learning (MSL). MSL was impaired, relative to sham, by both anodal and cathodal prefrontal stimulation.

The authors suggest their findings to indicate that the dorsolateral prefrontal cortex is involved in initial skill acquisition, through its role in working memory.

This study addresses an important topic, the functional anatomy of explicit MSL. I believe that the decision for offline stimulation is not optimal, but the authors have justified their choice. The results may hint at a role for left dorsolateral prefrontal cortex in the late part of the early phase of MSL.

I have 1 comment for consideration by the authors:

1) Although the experiments have been carefully designed with a view of the hypothesis of directionally specific stimulation-effects, the design has worked less well with the present, partially unexpected results. Because stimulation effects were seen after both anodal and cathodal stimulation, the present study lacks evidence of spatial specificity, i.e. of a negative stimulation site. This is a serious limitation which should be addressed best by adding new experiments targeting a brain area that is located outside the cerebello-thalamo-prefrontal circuit which the authors hypothesize is implicated in early MSL.

Minor:

1) Table 1. Please indicate units of current intensities in the table or its legend .

2) Could Figs 3 and 4; and Figs 5 and 6 be combined to 1 figure each?

Reviewer #2: Ballard  et al. conducted a straight forward study testing stimulation effects on motor learning. There are many positive aspects in this study, however it is not clear what is its theoretical contribution.My main comment is that there is no theoretical novelty in this study, apart the technical novelty of using HD tDCS in a well-studied task. Considering the weak results (which might reflect a small number of participants, considering the large drop-out rate). The lack of apriori power calculations - as long as power calculations were not conducted, means null results cannot be justified.Personally I think HD-tDCS is not a serious direction since electrode size is 10 times bigger than the target ROI - tDCS has poor spatial resolution, at least until new tiny electrodes will be used.

The lack of differences between anodal and cathodal effects just replicates the predictions of Shilo and Lavidor (2019, EBR), where 2 mA stimulation for 20 minutes reverse the direction of cathodal stimulation (as was first reported for MEP by Batsikadze et al., 2013).What is the purpose of Table 2 if it does not compare sensations between active and sham sessions? The analysis model requires justification - the selection of 3 learning phases seems posthoc. Motor learning paradigm (and the present one is similar to the implicit paradigms, apart from marking R and S on the sequences), require first to analyze the experimental conditions by the main variable of the paradigm, e.g. was the block random or repeated. The fact that the repeated sequences were not really repeated but differed from each other makes the manipulation weak, thus minimizing the expected differences between R and S. No wonder accuracy was low (that is the natural accuracy before filtering out 20% of subjects due to poor accuracy - I guess it might reflect the difficulty level of the procedure, where learning was limited as the repeated sequences were not identical)why learning phase was analyzed for accuracy but block type (R or S) for RT?

6. PLOS authors have the option to publish the peer review history of their article (what does this mean?). If published, this will include your full peer review and any attached files.

Reviewer #1: No

Reviewer #2: No

---

## [Author Response · Author response to Decision Letter 0]

10 Dec 2020

Editor:

As you will see below, the experts who have reviewed your paper have expressed significant concerns regarding the methodology and the research design. In particular, both Reviewers were concerned about the lack of specificity between anodal and cathodal stimulation. The issue of spatial selectivity is also a significant issue. I share their concerns and ask you to consider adding an experiment to address the spatial selectivity issue. There are also other considerations raised by Reviewer 2 regarding the statistical analysis that will require your attention.

We appreciate the opportunity to revise and resubmit the manuscript. We feel that the points raised by both the reviewers and the editor have helped to significantly improve our manuscript and its readability. 

We have ensured that the manuscript meets PLOS ONE’s style requirements, including those for file naming.

 2. We note that Figure 2 includes an image of a participant in the study. 

We thank the editor for requesting formal consent from the subject presented in Figure 2. In order to avoid this issue entirely, we have replaced this portion of Figure 2 with an image that does not contain a human subject. The source of this image is now disclosed in the caption of Figure 2. 

Reviewer #1:

The authors stimulated the dorsolateral prefrontal cortex by high-definition transcranial direct current stimulation to study its role in explicit motor sequence learning (MSL). MSL was impaired, relative to sham, by both anodal and cathodal prefrontal stimulation.

The authors suggest their findings to indicate that the dorsolateral prefrontal cortex is involved in initial skill acquisition, through its role in working memory.

This study addresses an important topic, the functional anatomy of explicit MSL. I believe that the decision for offline stimulation is not optimal, but the authors have justified their choice. The results may hint at a role for left dorsolateral prefrontal cortex in the late part of the early phase of MSL.

I have 1 comment for consideration by the authors:

3. Although the experiments have been carefully designed with a view of the hypothesis of directionally specific stimulation-effects, the design has worked less well with the present, partially unexpected results. Because stimulation effects were seen after both anodal and cathodal stimulation, the present study lacks evidence of spatial specificity, i.e. of a negative stimulation site. This is a serious limitation which should be addressed best by adding new experiments targeting a brain area that is located outside the cerebello-thalamo-prefrontal circuit which the authors hypothesize is implicated in early MSL.

We believe that our inclusion of a sham condition for all subjects in both the anodal and cathodal groups already addresses spatial specificity to a sufficient degree. The sham condition allows us to use performance after placebo stimulation as a baseline measure, which we can then compare with performance after active (anodal or cathodal) stimulation to the same area. As such, our experiments have been purposefully designed to account for location effects and distinguish between stimulation types.

Further, though it may be informative in certain ways, the inclusion of a negative stimulation site will complicate the interpretation of our results. Stimulating another brain area, outside of the cerebello-thalamo-prefrontal network, could cause alternative effects on performance, depending on the area being stimulated. For instance, if we chose to use the temporal lobe as a negative stimulation site, we could cause unwanted effects on sequence learning performance as this area is involved in explicit memory. One could also imagine similar potential effects for areas in the parietal lobe, or orbitofrontal cortex. Ultimately, stimulating any brain area outside of our network of interest could produce some sort of influence on sequence learning performance, as this task involves several unique aspects of cognition and motor function. This investigation was specifically aimed at exploring the role of the cerebello-thalamo-prefrontal network in explicit sequence learning; thus, we believe that involving other brain areas will obscure results and is outside the scope of this particular project.

Finally, due to COVID-19 safety regulations, data collection for human subjects research is currently limited at our institution. This restriction, along with the typical amount of time needed to complete a human subjects study of this nature, will cause significant delays for this work if an additional experiment is conducted. Given the rising number of COVID-19 cases in our local area, human subjects research may be more severely impacted in the near future. Thus, the addition of a third experiment would considerably impede the timely dissemination of results. While we agree that future work would benefit from the inclusion of an additional stimulation site, at this point, we feel that these results are interesting and important enough on their own, and the potential delay associated with an additional experiment is not worth what we would see as minimal scientific gain.

Minor:

4. Table 1. Please indicate units of current intensities in the table or its legend.

The units of measurement for current intensity (mA) have been added to the caption of Table 1.

5. Could Figs 3 and 4; and Figs 5 and 6 be combined to 1 figure each?

We thank the reviewer for their suggestion regarding Figures 3-6. For easier visualization, we have presented Figures 3 and 4 together (now labeled Figure 3) and Figures 5 and 6 together (now labeled Figure 4). Each pair of figures is presented in one panel with two sections, A and B. This is how we have interpreted the reviewer’s comment, and we would like to keep all four figures in the manuscript as each provides important information. However, we have consolidated these figures, rather than presenting each separately, so that they can be directly compared with one another.

Reviewer #2: 

Ballard et al. conducted a straight forward study testing stimulation effects on motor learning. There are many positive aspects in this study, however it is not clear what is its theoretical contribution.

6. My main comment is that there is no theoretical novelty in this study, apart the technical novelty of using HD tDCS in a well-studied task. Considering the weak results (which might reflect a small number of participants, considering the large drop-out rate). The lack of apriori power calculations - as long as power calculations were not conducted, means null results cannot be justified.

We understand the limitations of tDCS, and we agree that spatial resolution of stimulation is a concern, given the size of the electrodes relative to our targeted brain region. However, we believe that the novelty of using HD-tDCS, a cutting-edge method for non-invasive brain stimulation, is important in and of itself for advancing neuromodulation research. As indicated in lines 108-111 of the manuscript, this method has not been adequately explored with particular respect to the influence of prefrontal stimulation on explicit motor sequence learning. As HD-tDCS achieves improved targeting, a reevaluation of the existing knowledge on brain stimulation effects (formed from work employing the more traditional, less precise 2-pad method) is critical in improving our understanding of the neural mechanisms underlying sequence learning. 

However, speaking more directly to the theoretical novelty of our results, we feel that they contribute useful insight on the effects of different prefrontal stimulation types (anodal vs. cathodal) on behavior. The common approach in tDCS research is to attribute improved performance to excitation with anodal stimulation and worsened performance to inhibition with cathodal stimulation, but this theory is being challenged with recent advances in the field, such as HD-tDCS. Several studies have presented evidence against this standard belief, and we believe that our results add to the growing body of literature that supports this notion. We feel that the conclusions drawn from this investigation valuably contribute to the ongoing conversations regarding the particular effects of tDCS on behavior, suggesting that the influence of stimulation polarity (anodal vs. cathodal) may not be as simple as we thought. Given the complicated nature of this methodology, additional evidence is necessary to form solid conclusions. 

Finally, we would highlight that according to PLOS ONE’s criteria for publication, “theoretical novelty” is not listed. The complete list below (available here: https://journals.plos.org/plosone/s/journal-information) notes the importance of original research, newness of the results, and ethical standards, among others. We believe that our findings meet all of these criteria, and thanks to reviewer feedback, our manuscript has improved.

1. The study presents the results of original research.

2. Results reported have not been published elsewhere.

3. Experiments, statistics, and other analyses are performed to a high technical standard and are described in sufficient detail.

4. Conclusions are presented in an appropriate fashion and are supported by the data.

5. The article is presented in an intelligible fashion and is written in standard English.

6. The research meets all applicable standards for the ethics of experimentation and research integrity.

7. The article adheres to appropriate reporting guidelines and community standards for data availability.

We appreciate the reviewer’s attention to this matter. To more openly convey the theoretical novelty of our investigation, elaborating on the points discussed above, we have added wording to both the Discussion and Conclusion sections of the manuscript.

7. Personally I think HD-tDCS is not a serious direction since electrode size is 10 times bigger than the target ROI - tDCS has poor spatial resolution, at least until new tiny electrodes will be used.

Regarding spatial resolution, we agree with the reviewer that tDCS is not ideal, but this caveat is mitigated by our use of the more advanced HD-tDCS technique. This method achieves increased focality and spatial resolution, compared to the traditional 2-pad method, making it an important technique to further explore. Though there are still issues to work out with the methodology of tDCS as a whole, given this issue related to spatial resolution, we have presented work that takes advantage of the most optimal approach currently available by using the new and improved HD-tDCS technique. We have addressed this point in lines 106-108 and lines 441-445 of the manuscript. We have also directly noted this caveat in lines 485-488 of the Limitations section.

8. The lack of differences between anodal and cathodal effects just replicates the predictions of Shilo and Lavidor (2019, EBR), where 2 mA stimulation for 20 minutes reverse the direction of cathodal stimulation (as was first reported for MEP by Batsikadze et al., 2013).What is the purpose of Table 2 if it does not compare sensations between active and sham sessions? 

We thank the reviewer for noting the lack of comparison between active and sham sessions with Table 2. However, the stimulation sensations questionnaire was only administered at the end of the second session for each subject, during debriefing (please see lines 243-245 of the manuscript). As such, we did not record separate responses for active and sham sessions. Subjects responded to the questionnaire based on their collective experience from both sessions. We intended to keep subjects blind to the stimulation condition during each session, especially to avoid disclosing the presence of a placebo condition, and delivered the sensations questionnaire in such a way that this aspect of our experimental design was protected. Responses were used to determine whether the sensations experienced during stimulation may have interfered with subjects’ ability to subsequently perform the task. The results we have presented in Table 2 suggest that sensations from stimulation were not considerably strong, or thereby distracting, across sessions. To clarify the purpose of this questionnaire and its administration, we have added wording to the Debriefing section of the manuscript.

9. The analysis model requires justification - the selection of 3 learning phases seems posthoc. 

We appreciate the reviewer’s comment regarding the analysis model and selection of learning phases. This particular aspect of our model was a priori (as indicated in line 304 of the manuscript), rather than post hoc. In our past work on the cerebellum, we used this same approach and adapted that here, as noted in our manuscript (see also Ballard et al., 2019). To make this explicitly clear, we have added additional wording to the Data Analysis section to specifically state that the grouping of sequence blocks into three learning phases was an a priori decision. Justification for this model has been described in lines 278-286.

10. Motor learning paradigm (and the present one is similar to the implicit paradigms, apart from marking R and S on the sequences), require first to analyze the experimental conditions by the main variable of the paradigm, e.g. was the block random or repeated. 

Regarding the reviewer’s comment on analyzing the main variable of the paradigm (random vs. repeated blocks), we feel that the model described in lines 342-357 of the manuscript addresses this aspect of our design. Reaction time was analyzed using a 2 x 2 x 2 model that included block type (random vs. sequence). This variable was analyzed within a larger ANOVA in order to reduce multiple comparisons and simplify analyses. Our reasoning for analyzing block type for reaction time only, and not accuracy, is described in response #12 below.

11. The fact that the repeated sequences were not really repeated but differed from each other makes the manipulation weak, thus minimizing the expected differences between R and S. No wonder accuracy was low (that is the natural accuracy before filtering out 20% of subjects due to poor accuracy - I guess it might reflect the difficulty level of the procedure, where learning was limited as the repeated sequences were not identical).

We appreciate the reviewer’s comment regarding the repeated sequences. However, the sequences were in fact repeated during the course of the task, though different sequences were used between sessions in order to account for practice effects. We apologize for this lack of clarity and have adjusted wording in the Methods section of the manuscript to ensure that the repeated nature of the sequence within a session is made abundantly clear. This task was intended to be explicit in order to investigate the effects of stimulation on skill acquisition, in particular. In this investigation, we were not interested in evaluating implicit or long-term sequence learning, and, therefore, used two different sequences between sessions, rather than the same sequence across both sessions. We have explained this aspect of our experimental design in lines 230-238 of the manuscript. The explicit nature of our paradigm was indeed achieved by preceding each block with either an "R" or an "S", but also by using two different sequences between sessions, allowing us to evaluate the initial acquisition of a sequential pattern of finger movements over the course of a brief paradigm.

Finally, it is notable, as the reviewer pointed out, that several participants were dropped due to poor accuracy. However, we suspect that this was not due to poor learning; rather, this was likely due to a lack of attention or engagement with the task. Participants that were removed from analysis for this reason had accuracy at 34% (cathodal group) and 33% (anodal group) - relative to chance (noted in lines 132-135 and 143-145 of the manuscript). As we were using an undergraduate participant pool, motivation levels varied greatly between participants in the sample, as they only needed to show up in order to receive course credit. We have added to the description of this point in the Limitations section of our manuscript.

12. Why learning phase was analyzed for accuracy but block type (R or S) for RT?

Regarding the reviewer’s comment on our decision to analyze block type by reaction time and learning phase by accuracy, we chose to use this particular approach as it most adequately represents our thought process in assessing these variables. We first looked at reaction time across both block types, in order to confirm that we successfully implemented an explicit learning paradigm. Higher reaction times during random blocks, compared to sequence blocks, is expected given the broader sequence learning literature, and our reaction time results support this notion. We were then specifically interested in sequence blocks, as our primary goal for this investigation was to evaluate the impact of stimulation on the acquisition of a sequential pattern of finger movements. Accuracy for sequence blocks is expected to change over the course of the task as subjects become more familiar with the repeated sequence, whereas accuracy during random blocks generally remains relatively consistent over time. We appreciate the reviewer’s attention to this detail, and we have added a description of our justification for the overarching statistical approach to the Data Analysis section of the manuscript.

---

## [Decision Letter · Decision Letter 1]

19 Jan 2021

PONE-D-20-29379R1

Using high-definition transcranial direct current stimulation to investigate the role of the dorsolateral prefrontal cortex in explicit sequence learning

PLOS ONE

Dear Dr. Ballard,

Thank you for submitting your manuscript to PLOS ONE. After careful consideration, we feel that it has merit but does not fully meet PLOS ONE’s publication criteria as it currently stands. Therefore, we invite you to submit a revised version of the manuscript that addresses the points raised during the review process.

As you will see below, the Reviewers were mostly satisfied with the revised version of your manuscript. Reviewer #2 asks that the lack of a proper control experiment to address spatial specificity is recognized explicitly as a limitation in the discussion. Please address this comment and submit a revised version. 

We look forward to receiving your revised manuscript.

Kind regards,

François Tremblay, PhD

Academic Editor

PLOS ONE

Reviewers' comments:

Reviewer's Responses to Questions

**Comments to the Author**

1. If the authors have adequately addressed your comments raised in a previous round of review and you feel that this manuscript is now acceptable for publication, you may indicate that here to bypass the “Comments to the Author” section, enter your conflict of interest statement in the “Confidential to Editor” section, and submit your "Accept" recommendation.

Reviewer #1: (No Response)

Reviewer #2: All comments have been addressed

2. Is the manuscript technically sound, and do the data support the conclusions?

Reviewer #1: Yes

Reviewer #2: Yes

3. Has the statistical analysis been performed appropriately and rigorously? 

Reviewer #1: Yes

Reviewer #2: Yes

4. Have the authors made all data underlying the findings in their manuscript fully available?

Reviewer #1: Yes

Reviewer #2: Yes

5. Is the manuscript presented in an intelligible fashion and written in standard English?

Reviewer #1: Yes

Reviewer #2: Yes

6. Review Comments to the Author

Reviewer #1: The authors have carefully considered the questions raised by the reviewers. Overall, their responses are satisfactory, given the current circumstances.

I requested to consider adding an experiment addressing the spatial specificity of the effect observed by anodal and cathodal stimulation of the DLPC. In response, the authors “believe that involving other brain areas will obscure results”. I remain skeptical about this. After all, application of high-density stimulation is highlighted as one novel aspect of the study. This would be pointless, if the advance on spatial resolution would not also be harvested. If the authors think that adding any brain area will have deleterious effects on sequence learning, the effect seen after DLPC stimulation does not support any specific contribution to a cerebellar-frontal network. However, I understand the limitations imposed by the current pandemic. Therefore, waiving an additional experiment could be acceptable if the limitations resulting from the waiver are adequately described. I suggest to address this limitation more explicitly.

Reviewer #2: No further comments to add. The manuscript can be accepted as it is.

7. PLOS authors have the option to publish the peer review history of their article (what does this mean?). If published, this will include your full peer review and any attached files.

Reviewer #1: No

Reviewer #2: No

---

## [Author Response · Author response to Decision Letter 1]

25 Jan 2021

Reviewer #1: The authors have carefully considered the questions raised by the reviewers. Overall, their responses are satisfactory, given the current circumstances. I requested to consider adding an experiment addressing the spatial specificity of the effect observed by anodal and cathodal stimulation of the DLPC. In response, the authors “believe that involving other brain areas will obscure results”. I remain skeptical about this. After all, application of high-density stimulation is highlighted as one novel aspect of the study. This would be pointless, if the advance on spatial resolution would not also be harvested. If the authors think that adding any brain area will have deleterious effects on sequence learning, the effect seen after DLPC stimulation does not support any specific contribution to a cerebellar-frontal network. However, I understand the limitations imposed by the current pandemic. Therefore, waiving an additional experiment could be acceptable if the limitations resulting from the waiver are adequately described. I suggest to address this limitation more explicitly.

Response: We thank the reviewer for their thorough consideration of this point. We have noted the resulting limitation in the discussion section of the revised manuscript.

---

## [Editor Report · Decision Letter 2]

27 Jan 2021

Using high-definition transcranial direct current stimulation to investigate the role of the dorsolateral prefrontal cortex in explicit sequence learning

PONE-D-20-29379R2

Dear Dr. Ballard,

We’re pleased to inform you that your manuscript has been judged scientifically suitable for publication and will be formally accepted for publication once it meets all outstanding technical requirements.

Kind regards,

François Tremblay, PhD

Academic Editor

PLOS ONE
---

## [Editor Report · Acceptance letter]

10 Mar 2021

PONE-D-20-29379R2 

Using high-definition transcranial direct current stimulation to investigate the role of the dorsolateral prefrontal cortex in explicit sequence learning 

Dear Dr. Ballard:

I'm pleased to inform you that your manuscript has been deemed suitable for publication in PLOS ONE. Congratulations! Your manuscript is now with our production department. 

Kind regards, 

on behalf of

Dr. François Tremblay 

Academic Editor

PLOS ONE